# Non-Invasive Serum Biomarkers for the Diagnosis of Cirrhosis in Patients with Autoimmune Hepatitis (AIH) and AIH-Primary Biliary Cholangitis Overlap Syndrome (AIH-PBC): Red Cell Distribution Width to Platelet Ratio (RPR) Yielded the Most Promising Result

**DOI:** 10.3390/diagnostics14030265

**Published:** 2024-01-25

**Authors:** Siwanon Nawalerspanya, Jarukit Tantipisit, Suraphon Assawasuwannakit, Apichat Kaewdech, Naichaya Chamroonkul, Pimsiri Sripongpun

**Affiliations:** 1Gastroenterology and Hepatology Unit, Division of Internal Medicine, Faculty of Medicine, Prince of Songkla University, Songkhla 90110, Thailand; khuan_31117@hotmail.com (S.N.); suraphon@g.swu.ac.th (S.A.); apichat.ka@psu.ac.th (A.K.); naichaya@gmail.com (N.C.); 2Department of Internal Medicine, Phaholponpayuhasena Hospital, Kanchanaburi 71000, Thailand; 3Department of Pathology, Faculty of Medicine, Prince of Songkla University, Songkhla 90110, Thailand; medew.jarukit@gmail.com; 4Department of Medicine, Panyananthaphikkhu Chonprathan Medical Center, Srinakharinwirot University, Nonthaburi 11120, Thailand

**Keywords:** autoimmune hepatitis, primary biliary cholangitis, overlap syndrome, fibrosis, cirrhosis, red cell distribution width to platelet ratio

## Abstract

Several serum biomarkers for fibrosis assessment have been proposed in various liver diseases, but in autoimmune hepatitis (AIH) or overlap with primary biliary cholangitis (PBC; AIH-PBC) patients, the data are scarce. This retrospective cross-sectional study was conducted to validate six non-invasive biomarkers in the diagnosis of cirrhosis (F4 fibrosis) in such patients. We included adult patients diagnosed with AIH or AIH-PBC overlap syndrome who underwent a liver biopsy between 2011 and 2021. Laboratory data were collected to calculate the following scores: red cell distribution width to platelet ratio (RPR), aspartate aminotransferase/platelet ratio index (APRI), Fibrosis-4 index (FIB-4), aspartate aminotransferase (AST) to alanine aminotransferase (ALT) ratio (AAR), neutrophil-to-lymphocyte ratio (NLR), and lymphocyte-to-platelet ratio (LPR). A total of 139 patients were eligible (111 AIH and 28 AIH-PBC). The prevalence of cirrhosis was 35.3% (36% in AIH and 32.1% in AIH-PBC). The AUROCs of the RPR, FIB-4, APRI, AAR, LPR, and NLR in all patients were 0.742, 0.724, 0.650, 0.640, 0.609, and 0.585, respectively. RPR was significantly superior to APRI, NLR, and LPR. Moreover, RPR showed the highest AUROC (0.915) in the overlap AIH-PBC subgroup. In conclusion, RPR yielded the highest diagnostic accuracy to predict cirrhosis in AIH and AIH-PBC overlap syndrome patients, while FIB-4 was considerably optimal.

## 1. Introduction

Autoimmune hepatitis (AIH) is a chronic, non-resolving immune-mediated inflammatory liver disease that affects patients of all ages and ethnicities, with a female preponderance. The diagnosis of AIH can be ascertained by a combination of clinical characteristics, laboratory findings, and liver histology. The diagnostic scoring systems of the International Autoimmune Hepatitis Group (IAIHG), both the revised scoring system (1999) and the simplified scoring system (2008), were affirmed for use in the diagnosis of AIH, according to the clinical practice guidelines from the European Association for the Study of the Liver (EASL) and the American Association for the Study of Liver Diseases (AASLD) [1,2].

Nonetheless, there might be more than one autoimmune liver disease feature in a single person, a so-called ‘overlap syndrome’. The most commonly found overlap syndrome is AIH and primary biliary cholangitis (or the previous nomenclature was primary biliary cirrhosis) in combination (AIH-PBC), and the Paris criteria are generally recommended for use in the diagnosis of such a condition [1,2,3,4]. Once diagnosed, the incidence of liver cirrhosis detected at initial presentation is about 30–80% [5,6] in patients with AIH, and in those without cirrhosis at the diagnosis, 10% developed cirrhosis during follow-up or even after biochemical remission is detected, which means life-long monitoring is needed in these patients [7]. In addition, in AIH-PBC overlap syndrome, patients tend to present with more advanced fibrosis and portal hypertension-related complications than in patients with AIH alone or conventional PBC alone [2,3,4].

Accordingly, fibrosis and cirrhosis assessments at first diagnosis and during the follow-up period are of clinical importance. Once cirrhosis is diagnosed, it triggers a cascade of cirrhosis care, for example, screening for gastro-esophageal varices, either by esophagogastroduodenoscopy (EGD) or using non-invasive methods such as transient elastography in combination with platelet counts as per BAVENO VI recommendation, and hepatocellular carcinoma surveillance every 6 months [8,9,10]. Moreover, for the decision of therapeutic options in autoimmune liver diseases, budesonide is generally not recommended in patients who already have cirrhosis, as it increases the risk of portal vein thrombosis and systemic side effects regarding portosystemic shunting [1]. Liver biopsy is still a reference standard for the evaluation of the fibrosis stage. Despite high diagnostic accuracy, liver biopsy is an invasive procedure, and it comes with a risk of severe life-threatening complications [5]. A study regarding percutaneous liver biopsy in AIH patients with cirrhosis found a complication rate of 5.8% with some serious bleeding, even in patients with Child–Pugh class A patients [11]. Thus, a liver biopsy is not a suitable tool for a dynamic evaluation of the liver fibrosis stage during the follow-up period [12,13]. And recently, a study concerning the role of a liver biopsy for the diagnosis of AIH in patients with typical laboratory features concluded that a liver biopsy might be unnecessary in patients with compatible clinical criteria [14]. Hence, the non-invasive liver fibrosis assessment methods should be used to avoid the complications of a liver biopsy and aid in dynamic fibrosis evaluation [13].

Since there is a gap in the knowledge concerning the high-performance non-invasive serum biomarkers for cirrhosis assessment in AIH and AIH-PBC overlap syndrome patients, in this study, we aim to evaluate the diagnostic performance of six non-patented serum biomarkers in the diagnosis of cirrhosis (stage 4 fibrosis) in AIH and AIH-PBC overlap syndrome patients.

## 2. Materials and Methods

We conducted a single-center retrospective cross-sectional study at our institute, which is a tertiary care university hospital in Thailand. The study protocol was approved by the Institutional Human Research Ethics Committee (HREC), Faculty of Medicine, Prince of Songkla University, Thailand (REC. 64-429-14-1). The study was conducted under the ethical guidelines of the 1975 Declaration of Helsinki.

### 2.1. Study Population

We included all adult patients (aged at least 18 years old) diagnosed with AIH or AIH-PBC overlap syndrome who underwent a liver biopsy at our center between 2011 and 2021 with available laboratory data to calculate serum biomarkers of interest. The exclusion criteria were as follows: (1) concomitant with hepatitis B virus (HBV), hepatitis C virus (HCV), significant alcohol consumption, or sclerosing cholangitis; (2) presence of the liver tumor at the time of the liver biopsy; (3) active infection, hematologic disease, or other inflammatory disease that may interfere with the results of the biomarkers of interest; and (4) inadequate tissue sampling from the liver biopsy for a re-evaluation.

### 2.2. Data Collection

All histopathological specimens were reviewed by a single pathologist who was blinded to clinical and laboratory data (JT). Baseline demographic data of the patients, e.g., age, sex, weight, height, underlying diseases, and laboratory data at the day of liver biopsy were obtained from the Hospital Information System (HIS) for the calculation of the non-invasive serum biomarker scores. Cirrhosis was defined by the presence of F4 fibrosis by METAVIR system on histopathological re-evaluation.

### 2.3. Non-Invasive Biomarkers of Interest

Aiming for the generalizability of the biomarkers’ utility in the real-world context, we focused only on non-patented biomarkers using only baseline demographic data and routine laboratory variables. The following scores were calculated and evaluated for the diagnostic performances in our study: red cell distribution width to platelet ratio (RPR), [15] aspartate aminotransferase/platelet ratio index (APRI), Fibrosis-4 index (FIB-4) [16], aspartate aminotransferase (AST) to alanine aminotransferase (ALT) ratio (AAR) [17], neutrophil-to-lymphocyte ratio (NLR) [18], and lymphocyte-to-platelet ratio (LPR) [17]. The formulas of the scores are as follows:RPR: RDW (%)/PLT count (10^9^/L).APRI: (AST (U/L)/ULN of AST)/PLT count (10^9^/L) × 100.FIB-4: (age (years) × AST (U/L))/((PLT count (10^9^/L) × (ALT (U/L))^1/2^).AAR: AST (U/L)/ALT (U/L) [16].NLR: Absolute neutrophil count (cells/μL)/Absolute lymphocyte count (cells/μL).LPR: Absolute lymphocyte count (cells/μL)/PLT count (10^9^/L).

## 3. Statistical Analysis

All statistical analyses were performed using R program version 4.1.0 (Vienna, Austria). Descriptive statistics were used for baseline demographic data. Quantitative measurements were shown as mean ± SD or median with interquartile range (IQR) according to the distribution of observed values. To compare the groups of patients with and without cirrhosis, Chi-square test for categorical variables and Wilcoxon rank-sum test or t-test for continuous variables were used for the analysis. The diagnostic performance of each biomarker was analyzed using areas under receiver operating curve (AUROCs) for the ability to discriminate between cirrhosis and non-cirrhosis. DeLong’s test was used for the comparison of AUROC. A *p*-value of <0.05 was considered statistically significant.

## 4. Results

### 4.1. Baseline Characteristics

From 1377 liver biopsy specimens of non-malignant liver tissue in the HIS during the study period, a total of 139 patients were eligible and included for analysis. Of those, 111 (79.9%) were AIH patients, and 28 (20.1%) were overlap AIH-PBC patients. The prevalence of cirrhosis was 35.3% overall (36% in AIH and 32.1% in AIH-PBC). In the cirrhosis group, Child–Turcotte–Pugh (CTP) scores were class A 57.1%, class B 36.7%, and class C 6.1%, with a median model for end-stage liver disease (MELD) score of 9 (IQR: 7.5,16). The mean age of the entire cohort was 55 years; however, the mean age of patients in the cirrhosis group was significantly higher than the non-cirrhosis group (57 vs. 52 years, *p* = 0.005), and 80.6% were female. No significant difference between the cirrhosis group and non-cirrhosis group in baseline characteristics such as body weight, body mass index (BMI), and medical co-morbidities except for a higher proportion of diabetes was found in the cirrhosis group (34.7%) compared to the other group (6.7%, *p* < 0.001), and around one third of patients in both groups were treatment-experienced before undergoing a liver biopsy, as shown in Table 1.

### 4.2. Liver Biopsy Complications

Liver biopsy complications were reported in four (2.9%) patients; three were immediate post-procedural bleeding; however, blood transfusion was required in only one patient. One patient experienced severe pain with desaturation following the procedure, which improved after receiving pain control and oxygen support. No fatal cases related to liver biopsy procedures feature in this study.

### 4.3. Laboratory Results

For the complete blood count (CBC) results, patients with cirrhosis had significantly lower hematocrit (Hct) levels (mean 34.1% vs. 36.7%, *p* = 0.012) and platelet counts (median 166 × 10^3^/μL vs. 235 × 10^3^/μL, *p* < 0.001) than those with non-cirrhosis. The red cell distribution width (RDW) was also significantly different between both groups (median 15.8 vs. 14.2 fL, *p* = 0.034), while mean corpuscular volume (MCV), white blood cell count, and lymphocyte proportion were comparable between groups. For liver chemistries, the only variable showing a statistically significant difference between those with and without cirrhosis was serum albumin level (median 3.6 vs. 4 g/dL, *p* = 0.001, respectively), whereas bilirubin, AST, ALT, alkaline phosphatase (ALP), and globulin level were not significantly different as shown in Table 1. The median values of non-invasive serum biomarkers of interest were significantly different between the cirrhosis and non-cirrhosis groups, as shown in Table 2.

### 4.4. Diagnostic Performance of Serum Biomarkers

In this study, we investigated six non-invasive serum biomarkers to predict cirrhosis in patients with AIH or overlap AIH-PBC patients, including APRI, FIB-4, AAR, NLR, LPR, and RPR, as mentioned earlier. In the entire cohort, the AUROCs of the biomarkers shown in descending order are as follows: RPR (0.742), FIB-4 (0.724), APRI (0.650), AAR (0.640), LPR (0.609), and NLR (0.585), respectively (Figure 1). RPR was significantly superior to APRI (*p* = 0.029), NLR (*p* = 0.023), and LPR (*p* = 0.016) for the diagnosis of cirrhosis status, while it was non-significantly higher than FIB-4 and AAR. For the subgroup of AIH-PBC overlap syndrome patients, the AUROC of the RPR was still the highest among all the evaluated biomarkers (0.915). The AUROCs of all biomarkers in the AIH and AIH-PBC subgroups are shown in Table 3.

As only RPR and FIB-4 demonstrated good diagnostic performance (AUROCs > 0.70), we then evaluated the sensitivity and specificity of RPR and FIB-4 at the cutoffs mentioned in prior studies for the diagnosis of cirrhosis (Table 4). According to Wang et al.’s study, the optimal cutoff values of RPR were 0.083, 0.084, and 0.127 in identifying significant fibrosis, advanced fibrosis, and cirrhosis in patients with autoimmune hepatitis, respectively; therefore, we use RPR at 0.127 for the cutoff in the diagnosis of cirrhosis in this study [15]. In the current cohort, at the cutoff levels of RPR (≥0.127) and FIB-4 (≥3.24) [12], the RPR showed a significantly higher specificity for ruling in cirrhosis than that in FIB-4 (93.33% vs. 66.67%, *p* < 0.001, respectively).

## 5. Discussion

Our study shows that in patients with biopsy-proven AIH and overlap AIH-PBC, the prevalence of F4 fibrosis or cirrhosis was around one-third, and among the currently available non-invasive biomarkers, RPR showed a good diagnostic ability to distinguish between the cirrhosis and non-cirrhosis stages at the AUROC of 0.742, which is better than other biomarkers. 

The epidemiology of AIH varies worldwide; the estimated prevalence is 15–25 cases/10^5^ persons in Europe [19], 4 cases/10^5^ persons in Singapore [20], and 2.5% of chronic hepatitis patients in Thailand [6]. The clinical manifestations of AIH are presented in a broad spectrum from asymptomatic (25–34%) to acute hepatitis (25–75%) to acute liver failure (3–6%). [1,2] In untreated patients, progression to liver fibrosis or cirrhosis with complications often occurs.

In this study, most of the patients with AIH and AIH-PBC overlap syndrome were female (80.6%), and the prevalence of biopsy-proven cirrhosis was 35.3%, which was similar to previous studies. [6,21] AIH was diagnosed in almost 80%, while overlap AIH-PBC patients were around one-fifth of the entire cohort. The baseline laboratory results also showed significantly lower platelet counts and albumin levels in AIH or overlap AIH-PBC patients with cirrhosis, as well as in general cirrhotic patients from other causes. 

Abdominal ultrasonography, although it is commonly used to diagnose the cirrhotic status of patients in routine clinical practice, the accuracy of an ultrasound in the diagnosis of cirrhosis has been reported to be only 64–79% [22]. Moreover, if signs of portal hypertension, e.g., splenomegaly, ascites, or collateral vessels, were not considered, using only liver parenchymal echogenicity or surface characteristics on ultrasound showed limited sensitivity and specificity to ascertain the diagnosis of cirrhosis [23]. 

Currently, non-invasive tests (NITs) for fibrosis stage assessment in liver diseases are the subject of extensive research. Device-assisted elastography, such as vibration-controlled transient elastography, is the most widely used and validated method with high performance for the diagnosis of cirrhosis, but it requires a dedicated device, which might not be easily accessible in general practice and requires an experienced operator [5,13]. Another point of concern is that the diagnostic performance of vibration-controlled transient elastography may be reduced in cases of narrow intercostal spaces, ascites, or the presence of inflammation, venous congestion, or obstructive cholestasis, which results in an overestimation of fibrosis [5,24]. The same concern was also observed in other methods, e.g., shear wave elastography or magnetic resonance elastography. Data from a large multicenter cohort study by Llovet et al. also indicate that liver stiffness measurement is not accurate to predict AIH cirrhosis in the long term, according to the decrease in necro-inflammation with liver remodeling or fibrosis resorption after immunosuppressive therapy, which may deteriorate the diagnostic yield [7].

Hence, in this study, we selected only non-patented serum biomarkers to be the NITs of interest in our study, owing to their wide availability, good reproducibility, high applicability, and lack of extra costs from routine clinical practice. We specifically focused on the diagnosis of F4 fibrosis rather than significant (≥F2) or advanced (≥F3) fibrosis in this study, as the main shift in management of AIH and overlap AIH-PBC patients occurs when patients have cirrhosis, not at the lower fibrosis stage; for example, such diseases should be treated regardless of the fibrosis stage, but having cirrhosis precludes the use of budesonide treatment, and it sets off further cirrhotic care bundle for the patients. In other liver diseases, the threshold for commencing treatment may start at F2 or higher. 

The APRI and FIB-4 scores are well-known NITs among hepatologists, and recently, LPR [17], AAR, NLR, and RPR [15,25,26,27,28,29,30,31,32] have been introduced to the field of liver fibrosis assessment. The diagnostic performances of these non-invasive serum markers in fibrosis assessment were mostly validated in HBV, HCV, or non-alcoholic fatty liver disease patients (NAFLD; or the most recently updated nomenclature of metabolic dysfunction-associated steatotic liver disease: MASLD). Nonetheless, when it comes to autoimmune liver diseases, the aforementioned biomarkers were examined to a considerably lesser extent. And previous limited data have indicated suboptimal to moderate discriminatory accuracy, hence precluding their recommendation for use in patients with AIH and PBC according to the current guidance [1,5]. In the more recent studies, Yuan X et al. [17] conducted an analysis of the LPR in patients with AIH in 2020 and reported that the LPR had a high diagnostic performance for liver cirrhosis, with an AUROC of 0.936. In contrast, Wang H. et al. [15] reported in the same year that the RPR demonstrated a statistically significant improvement over APRI and FIB-4 in the identification of cirrhosis among patients with AIH. The most recent study, published in 2022, reported on a meta-analysis examining the diagnostic accuracy of APRI and FIB-4 in predicting cirrhosis among 759 patients with AIH. The results indicated that both APRI and FIB-4 demonstrated modest diagnostic accuracy, with mean AUROC values of 0.644 and 0.732, respectively [12]. The existing body of data has not yielded a definitive outcome on the efficacy of NITs in individuals with AIH.

We evaluated all the aforementioned serum NITs in Thai AIH and overlap AIH-PBC patients. Our data showed that the AUROCs of those NITs range from 0.58 to 0.74, and RPR yielded the highest AUROCs for the diagnosis of cirrhosis, significantly better than APRI, NLR, and LPR, although it was not significantly different from FIB-4 and AAR. For the LPR, the AUROC was the lowest (0.585) in contrast to the study by Yuan X et al. [17], but the high diagnostic performance of the RPR was consistent with the Wang H. et al. study [15]. The difference in the results of LPR performance between our study and the study by Yuan X et al. [17] might be due to the difference in the study population. In the prior study, Yuan X et al. [17] compared AIH patients and healthy controls, whereas we studied the more homogenous group of only AIH and overlap AIH-PBC patients, in which we considered that this represented the exact population of interest in the real clinical situation. In the study by Wang H. et al. [15], although the population of the patients was almost the same as that in this study and showed concordant results, the novelty of our study was a remarkably high AUROC of RPR in overlap AIH-PBC patients. 

It is not entirely understood why RPR is superior to other NITs in diagnosing cirrhosis in patients with AIH. RPR consists of two routine laboratory variables, RDW and platelets, reported in daily clinical practice. RDW abnormality was hypothetically linked to cirrhosis as portal hypertension leads to hypersplenism and, consequently, a shorter survival of red blood cells (RBC) [29]. Additionally, the chronic inflammatory status produces proinflammatory cytokines that inhibit RBC maturation, and impaired iron metabolism and erythropoiesis in chronic inflammation may also contribute to the increased RDW [33]. Low platelet counts, on the other hand, are already well-known to be associated with liver fibrosis and cirrhosis owing to the impairment of thrombopoietin production and hypersplenism in such conditions. Not only in patients with AIH, RPR has also been validated in other liver diseases such as HBV, HCV, and PBC patients, as well as chronic hepatitis patients with mixed etiology [25,26,34,35] and yielded an AUROC of 0.71–0.9 in predicting cirrhosis. For the NLR, in 2018, Li X et al. demonstrated an AUROC of 0.637 for predicting advanced liver fibrosis in AIH patients; later, the study by Zeng T et al. also showed the same trend with an AUROC of 0.680 for the diagnosis of cirrhosis [27,28]. When it comes to NLR and LPR, both of them are indicators of the immune response to systemic inflammation [27,36] rather than reflecting the cirrhosis consequences; thus, this could be a possible explanation for the superiority of the RPR over these two biomarkers. 

Due to the exclusion of AIH-PBC overlap syndrome patients from most studies’ eligibility criteria, no validated serum markers for fibrosis assessment are currently available for this subgroup so far. This is the first study to provide novel information regarding non-invasive serum biomarkers for predicting cirrhosis in patients with AIH-PBC overlap syndrome. In this subgroup population, RPR still produced the highest AUROC (0.91), and in the same manner, followed by FIB-4.

The virtues of our study are that all patients had biopsy-proven AIH or overlap AIH-PBC, and the liver biopsy specimens were reviewed by a single pathologist who was blinded to the patients’ data. The laboratory data were also available in all patients on the same day of liver biopsy as per institutional protocol before the procedure, and they were well documented in the HIS. Additionally, the sample size in this study was quite large among the studies regarding AIH patients, and we did not include healthy controls for the comparison. We studied only in the homogenous group of patients with autoimmune liver diseases, reflecting the real clinical practice encountered for the differentiation between cirrhotic and non-cirrhotic AIH in both treatment-naïve and treatment-experienced patients. And lastly, this is the first study to evaluate the NITs in patients with AIH-PBC overlap syndrome.

Nonetheless, we acknowledge some limitations of the present study. Firstly, although the sample size was quite large among studies in AIH patients, the total sample size of 139 patients may not be high enough to demonstrate the difference in diagnostic performances among RPR and other NITs, especially FIB-4 and AAR. Moreover, our study was conducted in a single-center university hospital; thus, this may limit the generalizability of our findings. Secondly, the design of the study is retrospective; thus, potentially unrecognized confounders interfering with the laboratory results, e.g., concomitant other drugs or herbal use or patients’ co-morbidity status, which could affect the NIT results, may exist. However, we had already excluded patients with hepatocellular carcinoma or other liver tumors and those with active infection, inflammation or hematologic diseases in which there could be a clinically significant alteration of the laboratory results used in the calculation of non-invasive scores. Lastly, we didn’t have available transient elastography data in all patients to compare or combine with serum biomarkers to improve validation. Anyway, one of our purposes is to seek out simple tools for cirrhosis evaluation that are practical to use even in resource-limited areas.

In conclusion, we have learned from our cohort that RPR yielded the highest diagnostic accuracy to predict the cirrhosis status in AIH and especially in the subgroup of AIH-PBC overlap syndrome patients compared with other available NITs, while FIB-4 was considerably optimal. High RPR (≥0.127) showed a high specificity of >90% in the ascertainment of cirrhosis in these patients. Nonetheless, further prospective multicenter studies are needed to validate the usefulness of RPR and to provide strong evidence-based data before endorsing it in routine clinical practice guidelines.

## Figures and Tables

**Figure 1 diagnostics-14-00265-f001:**
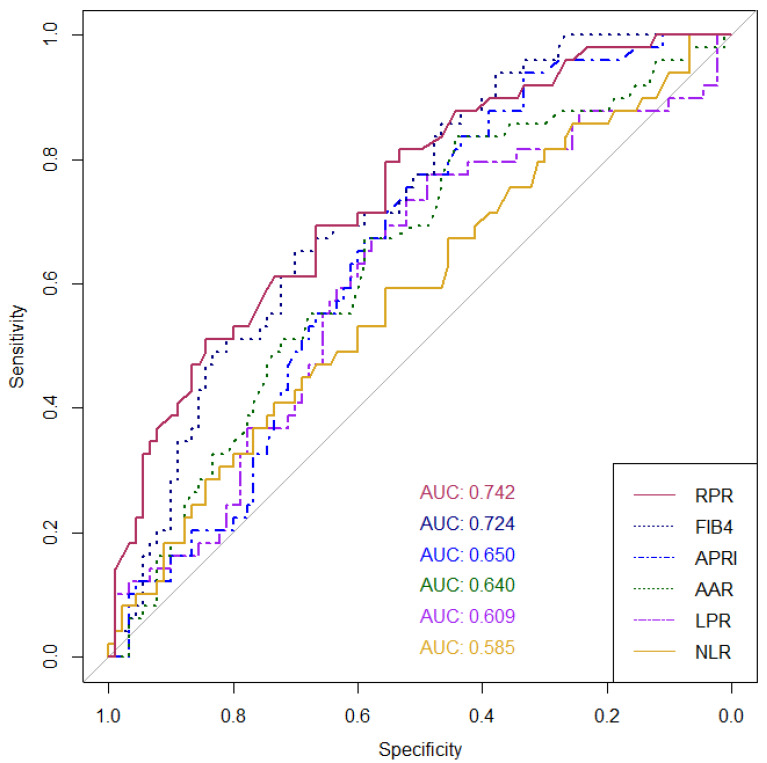
AUROCs for the ability of non-invasive serum biomarkers to discriminate between cirrhosis and non-cirrhosis in patients with autoimmune hepatitis (AIH) or overlap with primary biliary cholangitis (PBC; AIH-PBC overlap syndrome). Abbreviations: RPR, red cell distribution width to platelet ratio; APRI, aspartate aminotransferase/platelet ratio index: FIB-4, Fibrosis-4 index; AAR, aspartate aminotransferase to alanine aminotransferase ratio; NLR, neutrophil-to-lymphocyte ratio; and LPR, lymphocyte-to-platelet ratio.

**Table 1 diagnostics-14-00265-t001:** Baseline clinical characteristics between the non-cirrhosis group and the cirrhosis group in patients with autoimmune hepatitis (AIH) and overlap with primary biliary cholangitis (PBC; AIH-PBC overlap syndrome).

Variables	Total (*n* = 139)	Non-Cirrhosis (*n* = 90)	Cirrhosis (*n* = 49)	*p* Value
Age: mean (SD), years	55 (46.5,61)	52 (45,59.8)	57 (50,65)	0.005
Sex: female, N (%)	112 (80.6)	73 (81.1)	39 (79.6)	1
Weight: median (IQR), kg	58 (51,64.1)	58.6 (53.1,64)	57.8 (50.9,65)	0.595
BMI: median (IQR), kg/m^2^	23.7 (21.3,26.1)	23.8 (21.5,25.9)	23.6 (21.2,26.3)	0.899
**Final diagnosis**				0.870
AIH, N (%)	111 (79.9)	71 (78.9)	40 (81.6)	
AIH-PBC, N (%)	28 (20.1)	19 (21.1)	9 (18.4)	
IAIHG score, median (IQR)	17 (14,18)	17 (15,19)	15 (14,18)	0.046
Simplified criteria score, mean (IQR)	7 (6,7)	7 (6,7)	7 (5,7)	0.563
**Co-morbidities, *n* (%)**				
Diabetes mellitus	23 (16.5)	6 (6.7)	17 (34.7)	<0.001
Hypertension	35 (25.2)	18 (20)	17 (34.7)	0.089
Hyperlipidemia	43 (30.9)	29 (32.2)	14 (28.6)	0.800
Systemic lupus erythematosus	2 (1.4)	2 (2.2)	0	0.540
Thyroid disease	9 (6.5)	8 (8.9)	1 (2)	0.159
Other autoimmune disorders	10 (7.2)	8 (8.9)	2 (4.1)	0.494
**CBC parameters**				
Hct: mean (SD), %	35.8 (5.9)	36.7 (5.1)	34.1 (6.9)	0.012
MCV: median (IQR), fL	89.1 (80.1,93.3)	87.2 (80.1,92.3)	91.6 (80.3,95)	0.064
RDW: median (IQR), %	14.9 (13.2,17.6)	14.2 (12.9,17.2)	15.8 (14.3,17.8)	0.034
Platelet median (IQR), ×10³/uL	206 (146,274.5)	235 (172,307.8)	166 (121,218)	<0.001
WBC: median (IQR), ×10³/uL	6.2 (4.8,8.9)	6.5 (5,9.1)	5.7 (4.7,7.6)	0.357
PMN: mean (SD), %	56.8 (12.9)	55.1 (12.6)	59.8 (13)	0.042
Lymphocytes: mean (SD), %	32.1 (11.3)	33.4 (11.1)	29.9 (11.6)	0.082
**Liver chemistry**				
TB: median (IQR), mg/dL	1 (0.5,3)	0.9 (0.5,2.1)	1.3 (0.8,5)	0.069
DB: median (IQR), mg/dL	0.6 (0.2,2.7)	0.5 (0.2,1.9)	0.8 (0.3,4.2)	0.106
**AST: median (IQR), U/L**	95 (54.5,232)	89.5 (41.2,266)	106 (68,227)	0.178
**ALT: median (IQR), U/L**	94 (51,233)	102 (40.2,264.8)	82 (59,148)	0.783
ALP: median (IQR), U/L	132 (91,234)	123 (89.2,201)	155 (95,245)	0.182
ALB: mean (SD), g/dL	3.8 (3.5,4.3)	4 (3.6,4.4)	3.6 (3.1,4.1)	0.001
GLOB: median (IQR), g/dL	3.8 (3.2,4.5)	3.7 (3.1,4.3)	3.8 (3.3,4.5)	0.509
**Serologic evaluation, *n* (%)**				
ANA-positive (available *n* = 138)	108 (78.3)	71 (79.8)	37 (75.5)	0.715
SMA-positive (available *n* = 57)	17 (12.2)	13 (14.4)	4 (8.2)	0.301
AMA-positive (available *n* = 61)	20 (14.4)	17 (18.9)	3 (6.1)	0.014
IgG level: median (IQR), mg/dL (available *n* = 130)	2255.5 (1700,3060)	2271 (1700,3060)	2240 (1750,3060)	0.851
**Receive treatment before** **liver biopsy, N (%)**	43 (30.9)	29 (32.2)	14 (28.6)	0.800
Azathioprine, N (%)	21 (15.1)	15 (16.7)	6 (12.2)	0.654
Prednisolone, N (%)	30 (21.6)	19 (21.1)	11 (22.4)	0.974
UDCA, N (%)	10 (7.2)	7 (7.8)	3 (6.1)	0.986

Abbreviations: IAIHG, international autoimmune hepatitis group; HIV, human immunodeficiency virus; CBC, complete blood count; Hct, hematocrit; MCV, mean corpuscular volume; RDW, red cell distribution width; WBCs, white blood cells; PMNs, neutrophils; TB, total bilirubin; DB, direct bilirubin; AST, aspartate aminotransferase; ALT, alanine aminotransferase; ALP, alkaline phosphatase; ALB, albumin; GLB, globulin; ANA, antinuclear antibody; SMA, anti-smooth muscle antibody; AMA, anti-mitochondrial antibody; IgG, immunoglobulin G; UDCA, ursodeoxycholic acid.

**Table 2 diagnostics-14-00265-t002:** Comparisons of values of six non-invasive biomarkers between patients with and without cirrhosis (N = 139).

Biomarkers	Total (*n* = 139)	Non-Cirrhosis (*n* = 90)	Cirrhosis (*n* = 49)	*p* Value
RPR	0.076 (0.053,0.106)	0.062 (0.047,0.09)	0.101 (0.069,0.141)	<0.001
FIB-4	2.83 (1.295,6.045)	2.09 (1.002,4.71)	5.49 (2.29,8.81)	<0.001
APRI	1.41 (0.55,3.11	0.945 (0.363,2.73)	2.06 (1.07,3.26)	0.004
AAR	1.11 (0.77,1.555)	1.04 (0.742,1.375)	1.29 (1.01,1.77)	0.006
LPR	9.618 (7.116,13.78)	8.273 (6.697,13.104)	11.176 (8.241,13.866)	0.035
NLR	1.68 (1.185,2.675)	1.58 (1.128,2.33)	1.8 (1.32,3.13)	0.098

**Table 3 diagnostics-14-00265-t003:** Comparison of the AUROCs of non-invasive serum biomarkers to discriminate between cirrhosis and non-cirrhosis in patients with autoimmune hepatitis (AIH) or overlap with primary biliary cholangitis (PBC; AIH-PBC overlap syndrome).

Non-Invasive Serum Biomarkers	AUROC (95% Confidence Interval)		
	AIH	AIH-PBC	Entire Cohort
RPR	0.685 (0.582–0.789)	0.915 (0.812–1.000)	0.742 (0.658–0.827)
FIB-4	0.689 (0.590–0.790)	0.819 (0.657–0.981)	0.724 (0.640–0.809)
APRI	0.626 (0.521–0.730)	0.731 (0.537–0.925)	0.650 * (0.559–0.741)
AAR	0.601 (0.492–0.710)	0.784 (0.578–0.989)	0.640 (0.545–0.736)
LPR	0.601 (0.490–0.713)	0.626 (0.379–0.872)	0.608 * (0.509–0.709)
NLR	0.561 (0.450–0.672)	0.699 (0.452–0.946)	0.585 * (0.485–0.685)

Abbreviations: RPR, red cell distribution width to platelet ratio; APRI, aspartate aminotransferase/platelet ratio index; FIB-4, Fibrosis-4 index; AAR, aspartate aminotransferase to alanine aminotransferase ratio; NLR, neutrophil-to-lymphocyte ratio; and LPR, lymphocyte-to-platelet ratio. * *p* < 0.05 compared with RPR as a reference.

**Table 4 diagnostics-14-00265-t004:** Sensitivities and specificities of RPR and FIB-4 in the diagnosis of cirrhosis at a given cutoffs for an entire cohort.

Non-Invasive Serum Biomarkers	Cutoff Level	Sensitivity	*p*-Value	Specificity	*p*-Value
RPR	≥0.127	34.69%	ref	93.33%	ref
FIB-4	≥3.24	65.31%	<0.001	66.67%	<0.001

Abbreviations: RPR, red cell distribution width to platelet ratio; FIB-4, Fibrosis-4 index.

## Data Availability

The datasets generated and/or analyzed during the current study are not publicly available due to personal health information privacy policy and ethical restrictions but deidentification data are available from the corresponding author on reasonable request.

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
