# Peer review of "Non-Invasive Serum Biomarkers for the Diagnosis of Cirrhosis in Patients with Autoimmune Hepatitis (AIH) and AIH-Primary Biliary Cholangitis Overlap Syndrome (AIH-PBC): Red Cell Distribution Width to Platelet Ratio (RPR) Yielded the Most Promising Result"

_diagnostics, 2024, doi:10.3390/diagnostics14030265_

Round 1

Reviewer 1 Report

Comments and Suggestions for Authors

This study retrospectively describes the usefulness of no less than 6 biomarkers for a diagnosis of cirrhosis in AIH (and even a subgroup of  AIH-PBC)

My comments:

The group studied should be homogenous; i.e., include AIH alone not the overlap subgroup

To describe 6 biomarkers in 111 patients is too much

How was the RPR >0.127 cutoff chosen? The mentioned reference (19) describes APRI, not RPR?

I suggest you use this cohort as a learning cohort – and next use a future prospective validation cohort to check for the usefulness of RPR. That will increase the strength of the study considerably

Reviewer 2 Report

Comments and Suggestions for Authors

It is an interesting manuscript describing the potential role of RPR in the diagnosis of cirrhosis in the course of AIH and AIH/PBC overlapping syndrome. Could the authors enumerate more studies in the area of hepatology exploring the role of RPR? Except autoimmune liver disorders, which diseases were shown to be related to this marker, so far? Please, include more information on NLR in the discussion. How could the better diagnostic accuracy of RPR (in comparison with PLR and NLR) be explained in the context of autoimmunity and hepatology in general?

Reviewer 3 Report

Comments and Suggestions for Authors

Though the Paper has been written well , the limitations of the study should also be discussed because in clinical practice it is possible to                            
evaluate any patient with Autoimmune Hepatitis /PBC or overlap syndrome for the presence or absence of cirrhosis on the basis of imagine techniques and evidence of portal hypertension . In these situations , Serum biomarkers have very little to no role . 

Round 2

Reviewer 1 Report

Comments and Suggestions for Authors

My comments:

How was the RPR >0.127 cutoff chosen? Ref 15 is now given. This was a study that predicted advanced fibrosis (F3 and F4) and the RPR was given as lnRPR? Can you expand on this?

I still suggest you use this cohort as a learning cohort – make a much stronger future prospective validation cohort to check for the usefulness of RPR.

Round 3

Reviewer 1 Report

Comments and Suggestions for Authors

The questions raised have been answered well. Overall, I still recommend rejection - but I will of course leave it up to the editors to decide